# Effects of *Lactobacillus plantarum* Q180 on Postprandial Lipid Levels and Intestinal Environment: A Double-Blind, Randomized, Placebo-Controlled, Parallel Trial

**DOI:** 10.3390/nu12010255

**Published:** 2020-01-19

**Authors:** Ye Eun Park, Min Seo Kim, Kyung Won Shim, You-Il Kim, Jaeryang Chu, Byoung-Kook Kim, In Suk Choi, Ji Yeon Kim

**Affiliations:** 1Department of Food Science and Technology, Seoul National University of Science and Technology, Seoul 01811, Korea; nice0055_@naver.com (Y.E.P.); muckms@naver.com (M.S.K.); 2Department of Family Medicine, Ewha Womans University College of Medicine, Mokdong Hospital, Seoul 07985, Korea; ewhashim@ewha.ac.kr; 3Probiotics Research Laboratory, CKDBiO Research Institute, Ansan-si, Gyeonggi-do 15604, Korea; yikim@ckdbio.com (Y.-I.K.); chujr@ckdbio.com (J.C.); bkkim@ckdbio.com (B.-K.K.); inschoi6@ckdbio.com (I.S.C.)

**Keywords:** postprandial lipids, intestinal microbiota, intestinal microbiota metabolites, correlation, lipid mechanism, LPQ180, probiotics

## Abstract

Probiotics can improve the intestinal environment by enhancing beneficial bacteria to potentially regulate lipid levels; however, the underlying mechanisms remain unclear. The aim of this study was to investigate the effect of *Lactobacillus plantarum* Q180 (LPQ180) on postprandial lipid metabolism and the intestinal microbiome environment from a clinical perspective. A double-blind, randomized, placebo-controlled study was conducted including 70 participants of both sexes, 20 years of age and older, with healthy blood triacylglyceride (TG) levels below 200 mg/dL. Treatment with LPQ180 for 12 weeks significantly decreased LDL-cholesterol (*p* = 0.042) and apolipoprotein (Apo)B-100 (*p* = 0.003) levels, and decreased postprandial maximum concentrations (C_max_) and areas under the curve (AUC) of TG, chylomicron TG, ApoB-48, and ApoB-100. LPQ180 treatment significantly decreased total indole and phenol levels (*p* = 0.019). In addition, there was a negative correlation between baseline microbiota abundance and lipid marker change, which was negatively correlated with metabolites. This study suggests that LPQ180 might be developed as a functional ingredient to help maintain healthy postprandial lipid levels through modulating gut environment.

## 1. Introduction

Hyperlipidemia is a major risk factor for the development of serious cardiovascular diseases such as myocardial infarction, stroke, and atherosclerosis, especially when accompanied by chronic diseases such as hypertension and diabetes [1]. In particular, high blood triacylglyceride (TG) levels due to intake of a high-fat diet can contribute to the progression of arteriosclerosis [2,3]. Accordingly, there has been substantial research focused on identifying or developing functional food ingredients to control or prevent hypertriglyceridemia by managing TG levels.

Moreover, accumulating evidence points to the important role of intestinal microflora as an environmental factor affecting obesity and related diseases, including hyperlipidemia. An increase in harmful intestinal bacteria activates inflammatory pathways that affect lipid metabolism, leading to the development of obesity-related diseases [4]. With recognition of specific patterns of intestinal microbial metabolism associated with obesity, there has been increased interest in the development of probiotics that can reduce TG levels and lipids in the body. For example, consumption of shakes containing *Lactobacillus acidophilus*, *Bifidobacterium bifidum*, and fructo-oligosaccharide resulted in a significant increase in HDL-cholesterol and a reduction in glycemia in elderly people with type 2 diabetes mellitus [5]. In addition, consumption of yogurt containing *Lb. acidophilus* and *Bifidobacterium lactis* caused a significant decrease in cholesterol in patients with hypercholesterolemia [6], and consumption of milk fermented with *Lb. acidophilus* 145 and *Bifidobacterium longum* BB536 reduced LDL-cholesterol levels [7]. Furthermore, research on various probiotics on improving blood lipids has been preceded [8]. These beneficial effects were directly attributed to the improved intestinal microbial environment from ingestion of probiotics.

Among the available probiotics, *Lactobacillus plantarum* Q180 (LPQ180) is a safe plant-derived lactic acid bacterium that has been isolated from the feces of adults, which has demonstrated the ability to reduce blood TG levels both in vitro and in vivo animal experiments [9,10]. LPQ180 was confirmed to inhibit adipocyte differentiation of 3T3-L1 cells in vitro due to the reduction of C/EBPα and PPARγ levels [11]. In vivo experiments confirmed that obese mice fed a high-fat diet and supplemented with the LPQ180 strain exhibited reduced fat cell uptake and size [12]. However, there have been no human intervention trials about LPQ180 conducted to date.

Based on the hypothesis that LPQ180 has a positive effect on lipid metabolism by improving intestinal microbiota composition and related metabolites, we conducted a double-blind, randomized, controlled trial to determine the effects of LPQ180 consumption postprandial lipid responses after oral lipid tolerance test (OLTT) using TNO fat challenge formula [13] in men and women over 20 years of age with a healthy and slightly higher fasting TG levels (<200 mg/dL). In addition, the effect on fecal environments was evaluated. These findings can provide a scientific basis for the development of LPQ180 as a functional ingredient to maintain healthy postprandial lipid metabolism.

## 2. Materials and Methods

### 2.1. Subjects

Adults over the age of 20 who had blood TG levels less than 200 mg/dL were recruited to participate in this trial through posters and online advertising. All participants provided written consent and the protocol received approval from the Internal Review Board (IRB) of Ewha Woman’s University Mokdong Hospital (IRB No. EUMC 2016-06-040). This study was also registered on the International Clinical Trials Registry Platform of the WHO (ICTRP, No. KC T0002085). The exclusion criteria were as follows: Recommended Food Score (RFS) > 36 to exclude participants having healthy dietary habit [14,15]; body mass index (BMI) below 23 kg/m^2^ and over 35 kg/m^2^; regular intensity exercise (≥10 h/week); chronic smokers (≥20 cigarettes/day); alcoholism and those who consume 420 g alcohol per week; diagnosis of hyperlipidemia, diabetes, hypertension, liver disease, bowel disease, kidney disease, cardiovascular disease, cerebrovascular disease, pancreatitis, cancer, thyroid disease, dementia, Parkinson’s disease, depression, anorexia/bulimia, or multiple sclerosis; taking medication or herbal medicine at least two months before the first visit; taking any probiotics or prebiotics or sustained intake of dietary supplements that affect body fat lipid metabolism and bowel health within the previous month before the first visit; participation in another clinical research trial within three months before the first visit; and pregnant or lactating women

### 2.2. Study Design

The study was designed as a randomized, double-blind, placebo-controlled, and parallel trial. The selected subjects ingested LPQ180 or placebo capsules daily for 12 weeks after a two-week run-in period. Seventy subjects who met the selection criteria were randomly assigned to the placebo or LPQ180 group by block randomization method (Figure 1).

The LPQ180 group (n = 35) received two 400 mg capsules containing 4.0 × 10^9^ colony-forming units of LPQ180 per day. The placebo group (n = 35) took two capsules containing the same amount of maltodextrin, which did not contain any probiotic strains. Both products were obtained from CKD Bio Corp., Ltd. (Seoul, Korea). The quality of all products such as harmful microbial levels and heavy metal levels were also checked by CKD Bio Corp.

### 2.3. Sample Collection

Blood collection was performed after fasting for 12 h, and postprandial blood collection was performed at 0, 6, and 12 weeks after ingestion of a 500-mL TNO fat challenge formula (palm oil 60 g, dextrose 83.5 g, protein powder 20 g, and 320 mL water) along with the study capsules. Blood was collected using EDTA Vacutainer blood collection tubes (Becton Dickinson, Franklin Lakes, NJ, USA) and stored in a 1.5-mL Eppendorf tube at −80 °C. Plasma and red blood cells were separated from the blood by centrifugation (1500× *g*, 4 °C, 15 min). Chylomicrons were isolated by adding sodium chloride solution to the plasma [16].

The fecal samples were collected at Weeks 0 and 12 using commercial kit for stool sampling (Medi4you, Seoul, Korea). Participants sent stool samples with ice pack and immediately stored at −80 °C without any treatment until analysis.

### 2.4. Blood Lipid Biomarker in Plasma

Biochemical parameters were analyzed in ISO 15180 certificated lab (Green Cross Laboratory, Seoul, Korea) or Ewha Woman’s University Mokdong Hospital. TG and chylomicron (CM) TG plasma levels were analyzed using an enzyme analyzer (BioTek Instruments Inc., Winooski, VT, USA), and the results were detected with a triglyceride colorimetric assay kit (Cayman Chemical Company, Ann Arbor, MI, USA). Apolipoprotein (Apo)B-48 and ApoB-100 plasma levels were analyzed by LBIS Human Apo-48 ELISA Kit (Shibayagi Co., Ltd., Gunma, Japan) and Human Apolipoprotein B ELISA PRO kit (Mabtech AB, Stockholm, Sweden), respectively, and the absorbance was detected on a SpectraMax 190 and StatFax 2600 microplate reader (Molecular Devices, San Jose, CA, USA).

### 2.5. Intestinal Microbial Community Analysis

DNA extraction, library preparation, and next-generation sequencing (NGS) for microbiome analysis were all commissioned by Macrogen, Inc. (Seoul, Korea). In total, 134 fecal samples were collected and stored below −80 °C. To prepare the library, DNA was extracted from the fecal samples using a PowerSoil^®^ DNA Isolation Kit (MO BIO Laboratories, Carslbad, CA, USA). Thereafter, PCR was performed to amplify the V3-V4 region of 16S rRNA genes according to the Miseq sequencing Library Prep guide (Illumina Inc., San Diego, CA, USA) with the 16S V3-V4 primer 16S Amplicon PCR Forward Primer 5′-TCGTCGGCAGCGTCAGATGTGTATAAGAGACAGCCTACGGGNGGCWGCAG and 16S Amplicon PCR Reverse Primer 5′-GTCTCGTGGGCTCGGAGATGTGTATAAGAGACAGGACTACHVGGGTATCTAATCC.

A subsequent limited-cycle amplification step was then performed to add multiplexing indices and Illumina sequencing adapters. The final products were normalized and pooled, and the size of the libraries were verified using TapeStation DNA screentape D1000 (Agilent Technologies, Santa Clara, CA, USA). Sequencing was then performed on the HiSeq2500 platform (Illumina Inc.).

### 2.6. Intestinal Microbial Community Analysis

#### 2.6.1. Fecal Microbial metaBolites: Biogenic Amines, SCFAs, Indoles and Phenols

Analysis of the biogenic amines was performed with reference to the method described by Saarinen [17] using a Shiseido SI-2 HPLC system (Kyoto, Japan) equipped with a fluorescence detector. Separation was carried out on a Cadenza 5CD-C18 (5 µm, 250 mm × 3 mm) column equipped with a Cadenza 5CD-C18 Guard Cartridge (5 µm, 5 mm × 2 mm; Imtakt USA, Portland, OR, USA). Total biogenic amines were defined as the sum of agmatine, methylamine, ethylamine, pyrrolidine, dimethylamine, propylamine, tryptamine, butylamine, 2-phenyl ethylamine, putrescine, cadaverine, histamine, tryamine, spermidine, and sperimine (Sigma Aldrich Corporation. Ltd., St. Louis, MO, USA). Heptylamine was used as the internal standard (IS).

The short chain fatty acids (SCFAs) in the feces were determined according to the method described by Costabile et al. [18]. Unbranched SCFAs, including acetic acid, propionic acid, butyric acid, and valeric acid, and branched SCFAs, including, iso-butyric acid and iso-valeric acid (Sigma Aldrich Corporation), were used as external standard solutions. The sum of the branched and unbranched SCFAs was defined as the total SCFA content. 2-Ethyl butyric acid was used as the internal standard. The SCFAs were analyzed using GC with a flame ionization detector (FID) on an Agilent 6890N system (Agilent Technologies) equipped with a DB-FATWAX UI column (0.25 µm, 30 m × 0.25 mm; Agilent Technologies).

The indoles and phenols were analyzed according to the methods described by Flickinger et al. [19] and Park et al. [20]. GC-FID was performed essentially as described for the SCFA analysis, with the exception of the use of a DB-17 column (0.25 µm, 30 m × 0.25 mm; Agilent Technologies). Indole, phenol, *p*-cresol, and skatole were used as external standards in the range of 30–500 µmol/L, and 4-isopropylphenol (Sigma Aldrich Corporation) was used as the internal standard.

#### 2.6.2. Fecal Excretion Fat-Related Metabolites: Neutral Sterols and Bile Acids

Neutral sterols and bile acids were analyzed as described by Anita Fechner et al. [21] and Sylvia Keller [22] using an Agilent GC 7890B-5977B MSD GC-MS system (Agilent Technologies) equipped with a DB-5MS column (0.25 µm, 30 m × 0.25 mm). Total sterols were defined as the sum of cholesterol, coprostanol (Steraloids Inc., Newport, RI, USA), coprostanone (Carbosynth Ltd., Newbury, UK), cholestanol, cholestanone, and cholestenone, and 5α-cholestane was used as the internal standard. All reagents except for coprostanol and coprostanone were purchased from Sigma Aldrich Corporation

Sterol extracts were saponified by the addition of NaOH for bile acid analysis. The primary external standards were cholic acid and chenodeoxycholic acid, and the secondary external standards were iso-lithocholic acid (Steraloids Inc.), lithocholic acid, iso-deoxycholic acid (Steraloids Inc.), and deoxycholic acid; hyodeoxycholic acid was used as the internal standard. All reagents were purchased from Sigma-Aldrich Corporation unless otherwise indicated.

### 2.7. Statistical Analysis

The sample size was estimated to be 35 subjects per group to provide an 80% power of demonstrating a significant difference in TG levels, based on similar previous studies and a drop-out rate of 20% [23,24,25]. Cross-group comparisons of all randomized subjects were performed for intent of treatment (ITT) analysis. Correlation analysis was performed with a per-protocol (PP) analysis including only those who completed the study. Based on the normality assessment, the data were log transformed to obtain a normal distribution prior to analysis. Student’s *t*-test was used to compare the characteristics of the subjects at baseline for continuous variables and to compare microbial communities between treatment groups, whereas categorical variables were analyzed by the chi-square test. In all functional assessments, a linear mixed-effect model was used with group, time, and group-by-time interactions (group×time) as fixed-effect variables for analyzing changes at baseline from Weeks 0 to 12, and as random-effects variables to analyze the difference between groups according to the intake period. The area under the curve (AUC) was calculated using the trapezoidal rule for postprandial lipid and related metabolic parameters at both 0 and 12 weeks. Pearson’s correlation coefficients were used to determine the correlations among variables. Statistical analysis was performed using SAS version 9.4 (SAS Institute Inc., Cary, NC, USA). When a *p*-value was less than 0.05, the difference was considered to be statistically significant for all parameters.

## 3. Results

### 3.1. Subject Characteristics

Four subjects in the placebo group and four subjects in the LPQ180 group dropped out eight weeks into the trial, leaving a total of 62 subjects who completed the trial (Figure 1). One participant allocated in placebo group was excluded because of taking obesity medication and others withdrew their consents. The characteristics of the selected 70 participants are summarized in Table 1. The sex ratio of the subjects was the same in both groups. The mean age of the subjects was 48.3 ± 1.5 years, the mean TG level was 133.8 ± 7.5 mg/dL, and the mean total cholesterol (TC) level was 201.5 ± 24.1 mg/dL. There were no significant differences in baseline characteristics between the placebo and LPQ180 groups.

### 3.2. Changes in Fasting and Postprandial Blood Lipid Levels

The results of fasting and postprandial lipid and related metabolic indicators are summarized in Table 2. Only LDL-cholesterol (*p* = 0.042) and ApoB-100 (*p* = 0.003) plasma levels significantly decreased in the LPQ180 group compared to those in the placebo group after 12 weeks of capsule intake. However, postprandial TG responses showed significant changes according to LPQ180 administration after 12 weeks. Post-prandial lipid change biomarkers showed an increasing pattern in the placebo group, while LPQ180 supplementation group showed decreasing pattern, although no statistical significance comparing the values between 0 and 12 weeks (Table 2). The AUC of postprandial TG demonstrated a significant difference (*p* = 0.049) at 6 h, with a decreased tendency in the early period (AUC 0–4 h, *p* = 0.099; AUC 2–4 h, *p* = 0.067) and a significant decrease in the latter period after ingesting the fat formula (AUC 4–6 h, *p* = 0.023) when comparing between placebo and LPQ180 groups. The maximum concentration (C_max_) of postprandial TG also significantly decreased (*p* = 0.016), and the postprandial TG change in excursion tended to decrease (*p* = 0.075). The C_max_ of CM TG also significantly decreased (*p* = 0.020), with a tendency toward a decrease in postprandial AUC at 4–6 h (*p* = 0.058) and excursion (*p* = 0.067). However, the 4–6 h postprandial AUC of ApoB-48 significantly decreased (*p* = 0.036), with a decreasing tendency of the AUC at 0–6 h (*p* = 0.070) and C_max_ (*p* = 0.063). However, significant decreases in the ApoB-100 AUC were observed for the full 6 h after fat formula ingestion (*p* = 0.024) and in the early postprandial periods (AUC 0–2 h, *p* = 0.005; AUC 0–4 h, *p* = 0.011; AUC 2–4 h, *p* = 0.038). The C_max_ of ApoB-100 also significantly decreased in the postprandial period (*p* = 0.003). 

### 3.3. Intestinal Microbiome

The Shannon index, representing the diversity of intestinal flora determined by NGS, showed no significant changes in the placebo and LPQ180 groups (*p* = 0.183) (Figure 2A). Non-metric multidimensional scaling (NMDS) analysis of the intestinal microbiome based on the distance between groups at the species level also showed no significant changes in total microflora in the placebo and LPQ180 groups (Figure 2B). At the phylum level, Actinobacteria showed no significant change in abundance over the 12-week period (Figure 2C), whereas the operational taxonomic units (OTUs) of the Acterbacteria genus *Eggerthella* spp. tended to decrease in the LPQ180 group (*p* = 0.075), with no similar change observed in the placebo group (Figure 2D).

### 3.4. Changes of Intestinal Microbial Metabolites in Feces

Cross-group comparisons in the change in total intestinal microbial metabolites for 12 weeks (Table 3) showed that levels of total indole and phenol significantly reduced after 12 weeks of LPQ180 capsule ingestion compared to those of the placebo group (*p* = 0.019). However, there were no significant differences in the other metabolites between the two groups. Changes in individual values of metabolites are shown in Appendix A, and indole (*p* = 0.064) tended to decrease and p-cresol (*p* = 0.021) significantly decreased in the feces of the LPQ180 group compared to those of the placebo group.

### 3.5. Correlation between the Baseline Intestinal Microbiome and Changes of Blood Lipid-Related Markers and Metabolites

The correlations between changes in blood lipids or metabolites and baseline intestinal microbiome abundance are shown in Table 4, and the corresponding correlation coefficients between the baseline intestinal microflora and changes in lipid markers and metabolites are presented in Appendix A. Changes in TG, TC, ApoB, indole, and phenol levels in the LPQ180 group were negatively correlated with the baseline *Ruminococcus bromii* abundance. In addition, changes in TC and LDL-cholesterol were reduced in individuals with increased baseline levels of *Kineothrix alysoides* only in the LPQ180 group. Baseline *Barnesiella intestinihominis* abundance was negatively correlated with change in plasma TG level and showed a tendency to be negatively related to biogenic amines after 12 weeks of LPQ180 intake. The change in blood HDL-cholesterol concentration after LPQ180 intake was positively correlated with abundance of *Flavonifractor plautii*. Baseline *Streptococcus salivarius* and *Rothia mucilaginosa* levels were positively correlated with changes in SCFAs after 12 weeks of LPQ180 intake, and negatively correlated with primary bile acids. Conversely, no such correlations were observed in the placebo group for any taxa.

The correlations between baseline fecal metabolite levels and changes in lipid markers after 12 weeks are shown in Table 5, and the detailed correlation coefficients are presented in Appendix A. Baseline biogenic amine levels were negatively correlated with changes in TG AUC, CM TG C_max_, and ApoB-48 C_max_ in the LPQ180 group. In addition, higher baseline levels of indole and phenol resulted in a greater increase in HDL-cholesterol levels after 12 weeks of LPQ180 intake. Baseline neutral sterols tended to be negatively correlated with change in ApoB-100 in the LPQ180 group. However, similar to the microflora correlation analysis above, there were no significant correlations between baseline metabolite levels and lipid markers in the placebo group.

## 4. Discussion

There has been increasing interest in studying the ability of probiotics to maintain healthy blood lipid levels. In a 12-week study of 128 subjects, ingestion of strains *L. curvatus* HY7601 and *L. plantarum* KY1032 reduced TG levels, and increased the Apo AV and LDL particle size [26]. In another clinical study, low-fat yogurt made with *B. longum* was consumed by 32 subjects for four weeks, resulting in a reduction in TC levels [27]. In addition, 60 subjects that received an *L. plantarum* strain for 12 weeks showed reduced LDL-cholesterol levels [28]. These studies have motivated further research regarding the lipid improvement function of probiotics; however, elucidating the precise mechanism has been difficult. Based on promising results from in vitro studies and animal models, we sought to identify the effects of LPQ180 on the improvement of lipid levels in human subjects with relatively healthy or slightly higher TG levels and to establish the underlying mechanisms through analysis of intestinal microbial metabolites and microorganisms from fecal samples.

Previous clinical studies only focused on the effect of probiotics on plasma lipids, whereas the present study further highlights the effects of strain LPQ180 on the intestinal environment, as well as providing insight into mechanisms that further improve lipid levels under certain conditions. Intake of LPQ180 in the current study reduced LDL-cholesterol and ApoB-100 plasma levels, the major lipid markers in the blood on fasting, which concurred with results of previous studies [28,29,30]. Although only LDL-cholesterol and ApoB-100 levels were decreased by LPQ180 supplementation for 12 weeks, postprandial blood levels of lipid markers such as TG, ApoB-48, and ApoB-100 were also significantly decreased in the LPQ180 group compared to those in the placebo group. Previous studies have shown that CM TG levels in arterial plasma increased during the postprandial period with a peak at 240–300 min [31]. LPQ180 was screened using inhibitory effect for porcine pancreatic lipase activity [9]. Therefore, our results indicate that LPQ180 may help maintain healthy postprandial lipid metabolism, as the AUC value of CM TG was significantly reduced compared with that of the placebo group 4–6 h after TNO fat formula ingestion.

There is growing recognition of the importance and role of gut microorganisms in metabolic disorders such as obesity, metabolic syndrome, and diabetes [32,33,34]. In particular, a greater abundance of beneficial intestinal microflora is considered to be effective in improving intestinal immunity and overall intestinal health [35,36]. Although we did not find a significant effect of LPQ180 on gut microflora diversity, analysis of the microbial community at the genus level showed that abundance of *Eggerthella* spp. tended to decrease in the LPQ180 group after 12 weeks. According to Leo Lahti [37], Actinobacteria, the phylum to which *Eggerthella* belongs, plays a role in endogenous lipid metabolism and is positively correlated with plasma cholesterol. In addition, Fu et al. [38] analyzed the correlation between intestinal microflora and lipid level by age, sex, and host genetics, finding a significant positive correlation with *Eggerthella* spp. abundance. In the above study, the *Eggerthella* spp. has been shown to increase blood TG levels and reduce HDL-cholesterol. Therefore, the present results suggest that intake of LPQ180 may help improve blood lipids by reducing levels of *Eggerthella* spp.

Polysaccharides that cannot be digested by humans are decomposed by beneficial intestinal bacteria such as probiotics, resulting in the production of SCFAs as byproducts [39,40]. The accumulation of SCFAs reduces the pH of the intestinal environment, which consequently inhibits the growth of harmful bacteria in the colon and results in a suitable environment for beneficial intestinal microorganisms [41]. Conversely, harmful bacteria such as *Escherichia coli*, *Clostridium* sp., *Bacteroides fragilis*, *Enterococcus faecalis*, and *Proteus* sp. produce decay products such as ammonia, amines, indoles, and secondary bile acids [42,43,44,45]. Previous studies have demonstrated that high levels of branched SCFAs in hypercholesterolemic rats were correlated with a high prevalence of *Odoribacter* (Bacteroidetes) producing iso-valeric acid and with *Ruminococcus* (Firmicutes) requiring branched SCFAs [46]. Metabolite levels in the above in vivo study were associated with changes in intestinal microflora. The Shannon index is a kind of species diversity index. It is interpreted that the higher is the value, the higher is the diversity of microorganisms. In addition, in the case of NMDS analysis of the microbial community by sample, the more distinctly separated are the groups, the more it can be interpreted that the change of microflora occurred significantly. Unfortunately, the current study did not show a difference in microbiota diversity based on the count of OTUs. This lack of difference may be related to large individual variations because of the small number of subjects. We also did not find a significant effect of LPQ180 intake on changes in SCFAs. It seems that LPQ180 as probiotics have a limit to increase gut beneficial bacteria. However, the significant reduction of indole and phenols confirmed that intake of LPQ180 affected gut environments, positively.

Correlation analysis further confirmed the link between the intestinal microbial community and changes in the lipid profile induced by LPQ180. Reductions of TG, TC, ApoB, and LDL-cholesterol lipid markers were more notable in subjects with high baseline levels of gut bacteria *R. bromii* and *K. alysoides*, which produce SCFAs by decomposing indigestible components [47,48]. Subjects with a higher baseline level of *B. intestinihominis*, which has immunomodulatory effects by increasing inflammatory control cytokines such as IFNγ that have antiviral and antitumor properties [49], exhibited reduced levels of TG and biogenic amines. Subjects with higher baseline levels of *F. plautii*, which is known to degrade flavonoids in the intestine [50], also exhibited increased HDL-cholesterol levels after LPQ180 intake for 12 weeks. With respect to intestinal microbial metabolites, a higher baseline level of oral and intestinal commensal bacteria, including anti-inflammatory species *Streptococcus salivarius* [51], resulted in an increase in SCFAs and a decrease in primary bile acids in the LPQ180 group. In addition, subjects with a higher baseline level of *R. mucilaginosa*, which causes infection through induction of inflammatory marker cyclooxygenase-2 in hosts with compromised immunity [52], exhibited increased SCFAs and decreased primary bile acids. Moreover, correlation analysis demonstrated that LPQ180 significantly help to maintain postprandial lipid responses in subjects with high levels of metabolites such as biogenic amines and indole and phenols. No significant correlations of any of these biomarkers were detected in the placebo group, and higher indole and phenol levels were actually associated with increased changes in TG, VLDL, CM TG, and ApoB-48, demonstrating the opposite effects observed in the LPQ180 group.

## 5. Conclusions

LPQ180 ingestion ameliorated postprandial lipid metabolism and maintained a healthy intestinal environment. LPQ180 helped to maintain healthy postprandial lipid metabolisms in subjects with a higher level of enteric bacteria such as *R. bromii*, *K. alysoides*, *B. intestinihominis*, and *F. plautii*. In addition, increased baseline levels of these bacteria significantly increased the SCFA content after LPQ180 supplementation for 12 weeks. However, the results of this study alone are limited to explain the mechanism by which LPQ180 reduces postprandial lipid levels. This is because there was no statistical significance in gut microbiota levels and fating lipid variables by LPQ180 intake. This may be due to large deviations from the small number of subjects in each group and relatively healthy TG levels. To confirm the current correlation analysis, another clinical trial for monitoring the changes of postprandial lipid metabolism according to LPQ180 supplementation using participants having higher baseline levels of enteric bacteria such as *R. bromii*, *K. alysoides*, *B. intestinihominis*, and *F. plautii* or biogenic amines in feces might be needed.

## Figures and Tables

**Figure 1 nutrients-12-00255-f001:**
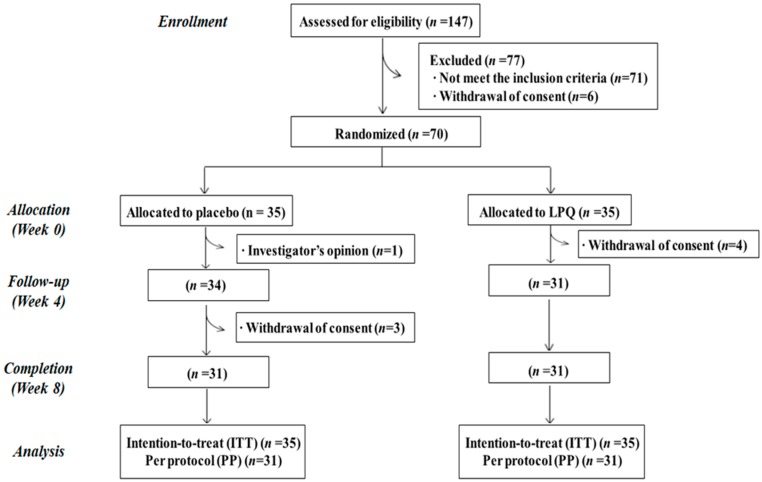
Flow chart of registered participants for the clinical trial.

**Figure 2 nutrients-12-00255-f002:**
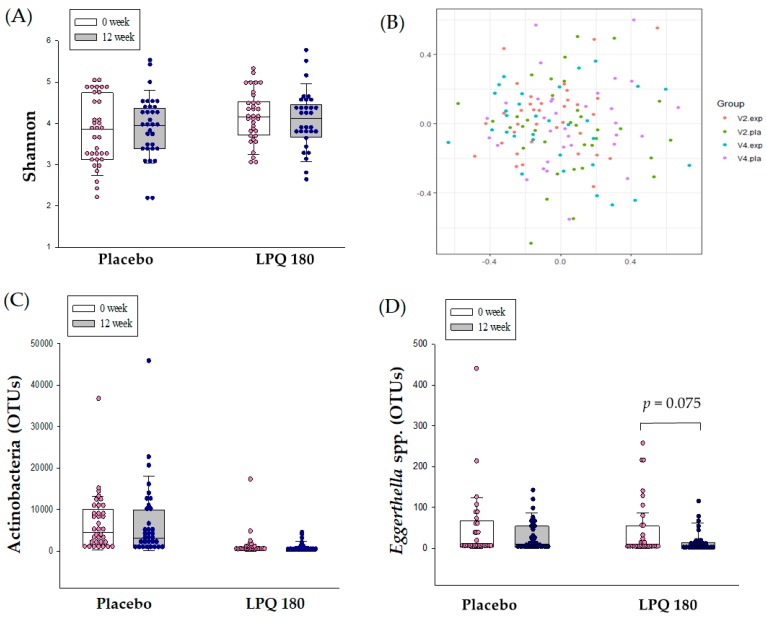
Changes in the intestinal microflora: (**A**) intestinal flora diversity represented by Shannon index (Alpha-diversity); (**B**) distance analysis between microbial communities by group (NMDS: Non-metric Multidimensional Scaling); (**C**) 12-week change in the placebo and LPQ180 groups of Actinobacteria, the upper level of *Eggerthella* spp.; and (**D**) microbial community significantly changing between Week 0 and Week 12 at genus level (*Eggerthella* spp.).

**Table 1 nutrients-12-00255-t001:** Baseline characteristics of the subjects ^1^.

Variables	Placebo (*n* = 35)	LPQ 180 (*n* = 35)	*p*-Value ^2^
Gender (male/female)	12/23	12/23	1.000
Age (year)	48.3 ± 13.2	48.3 ± 11.6	1.000
Menopause (Y/N/NA)	14/9/12	13/10/12	0.956
Postmenopausal period (month)	118.1 ± 101.2	114.8 ± 91.2	0.930
Body weight (kg)	68.9 ± 11.1	69.8 ± 11.2	0.745
BMI (kg/m^2^)	25.7 ± 2.5	26.4 ± 2.5	0.209
Waist circumference (cm)	85.6 ± 6.7	86.7 ± 7.5	0.493
Hip circumference (cm)	96.7 ± 4.8	97.1 ± 6.4	0.737
Alcohol drinker (Y/N)	20/15	15/20	0.232
Alcohol amount (g/week, per drinker)	44.9 ± 61.2	27.1 ± 26.3	0.254
Smoker (Y/N)	5/30	4/31	1.000
Smoking amount (cigarettes/day, per smoker)	9.0 ± 2.2	7.0 ± 4.2	0.390
Physical Activity (MET-min/week)	1335.8 ± 1150	1757.1 ± 2375	0.350
Blood lipid profiles			
TG (mg/dL)	134.2 ± 66.7	133.4 ± 58.9	0.958
TC (mg/dL)	206.3 ± 37.1	196.8 ± 41.6	0.316
LDL-C (mg/dL)	120.5 ± 31.3	114.4 ± 36.4	0.458
HDL-C (mg/dL)	50.1 ± 11.8	46.9 ± 11.4	0.253
VLDL-C (mg/dL)	26.8 ± 13.3	26.7 ± 11.8	0.958
Blood pressure			
SBP (mmHg)	120.0 ± 13.4	122.9 ± 15.1	0.410
DBP (mmHg)	71.7 ± 8.1	73.0 ± 10.2	0.543
RFS	23.5 ± 6.9	23.6 ± 6.8	0.944
MEDIFITCTS score	36.4 ± 22.2	38.6 ± 24.0	0.695
Dietary intake			
Energy (kcal/day)	1402.4 ± 429.5	1549.0 ± 377.5	0.156
Carbohydrate (g/day)	195.2 ± 74.9	226.2 ± 63.0	0.081
Protein (g/day)	59.9 ± 20.1	58.3 ± 17.2	0.729
Fat (g/day)	42.4 ± 17.7	45.6 ± 20.5	0.506
Sodium (mg/day)	3187.3 ± 1542	2995.6 ± 1041.4	0.564

^1^ Mean ± SD (all such values). LPQ180, *Lactobacillus plantarum* Q180; NA, not applicable; BMI, body mass index; RFS, recommended food score; MEDFICTS, meats, eggs, dairy, frying foods, in baked goods, convenience foods, table fats, snack; TG, triglyceride; TC, total cholesterol; LDL-C, low density lipoprotein cholesterol; HDL-C, high density lipoprotein cholesterol; VLDL-C, very low density lipoprotein cholesterol; SBP, systolic blood pressure; DBP, diastolic blood pressure. ^2^ Student’s *t*-test for continuous variables and Chi-square test for categorical variables were used to compare the difference between the groups.

**Table 2 nutrients-12-00255-t002:** Changes of summary valued of lipids after 12 weeks ^1^.

Variables	Placebo (*n* = 35)	LPQ 180 (*n* = 35)	*p*-Value ^2^
**Fasting Lipid**			
TG (mg/dL)	20.6 ± 9.9	−2.0 ± 9.9	0.243
TC (mg/dL)	1.6 ± 3.6	−8.1 ± 3.6	0.165
LDL-C (mg/dL)	13.8 ± 2.8	2.3 ± 3.0	0.042
HDL-C (mg/dL)	4.0 ± 1.1	3.1 ± 1.1	0.682
VLDL-C (mg/dL)	4.1 ± 2.0	−0.4 ± 2.0	0.243
Chylomicron TG (mg/dL)	38.9 ± 8.9	25.0 ± 9.5	0.442
ApoB-48 (ng/mL)	1.0 ± 4.7	−11.2 ± 5.0	0.202
ApoB-100 (ng/mL)	47.8 ± 13.8	−38.2 ± 14.6	0.003
**Postprandial lipid**			
TG			
AUC (mg min / dL)			
0–2 h	4112 ± 1458	−183 ± 1550	0.160
0–4 h	7821 ± 2729	−1655 ± 2901	0.099
0–6 h	11,714 ± 3808	−4171 ± 4049	0.049
2–4 h	3708 ± 1336	−1473 ± 1421	0.067
4–6 h	3893 ± 1326	−2516 ± 1410	0.023
Excursion (mg/dL)	19 ± 12	−26 ± 13	0.075
C_max_ (mg/dL)	53 ± 15	−26 ± 16	0.016
T_max_ (min)	−14 ± 11	16 ± 12	0.208
Chylomicron TG			
AUC (mg min / dL)			
0–2 h	5234 ± 1178	2756 ± 1253	0.314
0–4 h	9921 ± 2168	4781 ± 2305	0.257
0–6 h	14,744 ± 3014	5413 ± 3204	0.140
2–4 h	4687 ± 1035	2025 ± 1101	0.219
4–6 h	4823 ± 1045	632 ± 1111	0.058
Excursion (mg/dL)	7 ± 10	−33 ± 11	0.067
C_max_ (mg/dL)	51 ± 12	−7 ± 13	0.020
T_max_ (min)	−14 ± 12	−5 ± 13	0.735
ApoB-48			
AUC (ng min / mL)			
0–2 h	640 ± 509	−1017 ± 541	0.122
0–4 h	1515 ± 1164	−2320 ± 1238	0.117
0–6 h	2240 ± 1323	−2818 ± 1407	0.070
2–4 h	875 ± 656	−1304 ± 698	0.115
4–6 h	725 ± 274	−498 ± 292	0.036
Excursion (ng/mL)	12 ± 7	6 ± 7	0.672
C_max_ (ng/mL)	14 ± 5	−6 ± 5	0.063
T_max_ (min)	−51 ± 20	−10 ± 22	0.341
ApoB-100			
AUC (ng min / mL)			
0–2 h	5779 ± 1535	−3597 ± 1632	0.005
0–4 h	9047 ± 3046	−7560 ± 3239	0.011
0–6 h	10,212 ± 4492	−11,442 ± 4776	0.024
2–4 h	3269 ± 1637	−3963 ± 1740	0.038
4–6 h	1165 ± 1595	−3882 ± 1696	0.132
Excursion (ng/mL)	−14 ± 9	−3 ± 9	0.560
C_max_ (ng/mL)	45 ± 12	−35 ± 13	0.003
T_max_ (min)	−65 ± 29	−26 ± 31	0.519

^1^ LSmean ± SE (all values). LPQ 180, *Lactobacillus plantarum* Q180; TG, triacylglyceride; AUC, area under the curve; Excursion, difference between minimum and maximum concentration; C_max_, maximum concentration; T_max_, time to reach maximum concentration. ^2^ A linear mixed-effect model was used to analyze the effect of the group by time interaction for 12 weeks.

**Table 3 nutrients-12-00255-t003:** Comparison of intestinal microbial metabolites in feces for 12 weeks ^1^.

Variables	Placebo (*n* = 35)	LPQ 180 (*n* = 35)	*p*-Value ^2^
Total Biogenic amines(mg/g)	3.62 ± 2.91	−0.62 ± 3.01	0.459
Short chain fatty acid (µg/g)			
Unbranched SCFA	40.83 ± 28.95	−4.22 ± 30.57	0.433
Branched SCFA	4.09 ± 7.62	−3.39 ± 8.05	0.620
Total SCFA	44.93 ± 35.69	−7.61 ± 37.69	0.458
Total Indoles and phenols (µg/g)	44.27 ± 78.54	−334.49 ± 86.33	0.019
Total Neutral sterol (µg/g)	69.84 ± 39.65	65.20 ± 39.65	0.952
Bile acids (µg/g)			
Primary	−0.54 ± 0.50	0.70 ± 0.50	0.205
Secondary	36.23 ± 27.21	36.01 ± 27.21	0.997
Total bile acid	35.69 ± 27.20	36.70 ± 27.20	0.985

^1^ LS mean ± SE (all values). LPQ 180, *Lactobacillus plantarum* Q180. ^2^ A linear mixed-effect model was used to analyze the effect of the group by time interaction for 12 weeks.

**Table 4 nutrients-12-00255-t004:** Correlation between baseline levels of the intestinal microbiome and changes in blood lipid markers or fecal metabolites for 12 weeks ^1^.

Variables		Placebo (*n* = 31)		LPQ 180 (*n* = 31)	
		r ^2^	*p*	r ^2^	*p*
*Ruminococcus bromii*	Δ TG	0.118	0.689	−0.325	0.075
	Δ TC	0.119	0.523	−0.327	0.073
	Δ Apo B	0.032	0.864	−0.331	0.069
	Δ Indole and phenols	0.017	0.929	−0.447	0.012
*Kineothrix alysoides*	Δ TC	−0.012	0.950	−0.345	0.058
	Δ LDL-C	−0.106	0.571	−0.420	0.019
*Barnesiella intestinihominis*	Δ TG	0.040	0.892	−0.407	0.023
	Δ Biogenic amines	−0.056	0.766	−0.318	0.082
*Flavonifractor plautii*	Δ HDL-C	0.174	0.351	0.414	0.021
*Streptococcus salivarius*	Δ SCFA	0.163	0.382	0.508	0.004
	Δ primary bile acids	−0.228	0.218	−0.520	0.003
*Rothia mucilaginosa*	Δ SCFA	0.105	0.574	0.518	0.003
	Δ primary bile acids	−0.227	0.219	−0.556	0.001

^1^ LPQ 180, *Lactobacillus plantarum* Q180. ^2^ r: Pearson’s correlation coefficients.

**Table 5 nutrients-12-00255-t005:** Correlation between baseline levels of fecal metabolites and changes in blood lipid markers for 12 weeks ^1^.

Variables		Placebo (*n* = 31)		LPQ 180 (*n* = 31)	
		r ^2^	*p*	r ^2^	*p*
Biogenic amines	Δ TG AUC	−0.243	0.204	−0.348	0.104
	Δ CM TG C_max_	−0.326	0.104	−0.373	0.080
	Δ Apo B-48 C_max_	−0.156	0.428	−0.437	0.033
Indole and phenols	Δ HDL-C	0.219	0.237	0.407	0.023
Neutral sterol	Δ ApoB-100	−0.111	0.575	−0.308	0.092

^1^ LPQ 180, *Lactobacillus plantarum* Q180. ^2^ r: Pearson’s correlation coefficients.

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
