# Peer review of "Effects of Lactobacillus plantarum Q180 on Postprandial Lipid Levels and Intestinal Environment: A Double-Blind, Randomized, Placebo-Controlled, Parallel Trial"

_nutrients, 2020, doi:10.3390/nu12010255_

Round 1

Reviewer 1 Report

Overall it is very clear that you have gathered a lot of data and the English is very good. Mostly I could follow your rationale, logic and procedure throughout.

Introduction: Clear but I think that though the evidence is starting to grow in regards to triglycerides the absolute certainty that reducing triglycerides will reduce atherosclerosis is still not 100% there. There has been a recent RCT that is interesting but hypothesising that triglyceride reduction is important then selecting participants with low triglycerides sits a little uneasily. I think the conclusion of the introduction is also a bit too ambitious.

Methods: What is a recommended food score (is the MEDFICTS score you mention in the results)? Why did the investigator remove one participant - did they not meet the inclusion criteria? In regards to the capsules could you confirm that they contained what you think they did or is there some sort of accreditation you could quote in regards to CKD Bio corp to reassure us they will be of a quality appropriate for your study? When was the faeces collected? The analytical methods are explained in good detail. I assume the labs were commercial, as stated, or university or clinical hospital? Are the labs accredited e.g. according to ISO 15189? Did you do any power calculations before recruitment. I am not a statistical expert but in reading about all the methods done I wonder if there are enough samples to justify so many analyses and comparisons? 

Results: The table of differences shows those changes that were significant. In some cases the degree of rise wasn't much e.g. LDL. Why were the values increasing so much in the placebo group? Was the placebo tablet making lipids worse? In looking at the changes one thinks about whether the changes are clinically significant, which brings us back to the power calculation, what sort of change in triglyceride would you need to cause to have a meaningful effect on CVD? I am not sure we actually know that but it seems the effect is likely less than that required particularly if you account for biological and analytical variation. Microbial analysis is not my area of expertise. On Figure 2 I am afraid I don't fully understand. What do V2 and V4 and the colours in the 2 charts represent? Is this something to do with the primers used to analyse? You also talk about significantly different here but the p value is non-significant isn't it? I am now also a bit confused when you refer to table S1 - where is that (I have 3 supplemental figures but no table). The p value for Indole isn't significant and p-cresol is higher than the 0.019 you quote in the table for indoles and phenols together. I may be getting confused though. The number of comparisons again I find slightly concerning. Looking at the supplemental figures I wonder if those are actually best dispensed with so you can focus on a simple message otherwise I think you need to explain and justify this use of the data better. Does the effect of baseline characteristic on end effect mean that some people you would predict would respond better than others if you were using this therapeutically?

discussion: does the reduction in indoles etc really prove you have reduced harmful bacteria, should we more circumspect? The discussion certainly discusses much of the literature but some of it I wasn't sure how relevant it was to your actual data and I felt it lacked a critical appraisal of your data. 

All in all I think you have some excellent data that is mostly very well described. I think you should simplify it and need to review the validity of so many comparisons. The absolute changes seemed small and the inclusion criteria fierce so I think you also have to be cautious about what you can actually conclude. This is an early exploratory study which is fine but make sure you write it up as such. I am however limited by being primarily a lipid clinician with expertise in biochemistry so an expert in microbiome may be better able to evaluate some of your data and may conclude otherwise. Thank you.

Author Response

Thank you for your valuable comments. We have revised the manuscript based on all the comments in the reviewer.

Clear but I think that though the evidence is starting to grow in regards to triglycerides the absolute certainty that reducing triglycerides will reduce atherosclerosis is still not 100% there. There has been a recent RCT that is interesting but hypothesising that triglyceride reduction is important then selecting participants with low triglycerides sits a little uneasily. I think the conclusion of the introduction is also a bit too ambitious.

: Thank you for your valuable feedback. The current study was designed to monitoring postprandial responses for lipid parameters after LPQ180 supplementation. LPQ180 was screened using inhibitory activity of porcine pancreatic lipase and developed to use as a food supplement or functional food ingredient. Therefore, because of ethical and regulatory issue, we included relatively healthy populations. In this perspective, we revised our manuscript and title focused on the postprandial responses. Also, we toned down in the end of our abstract. (Title; Line 19-20, Page 1; Line 64-68, Page 2)

What is a recommended food score (is the MEDFICTS score you mention in the results)?

: The Recommended Food Score (RFS) has been developed to check dietary quality based on antioxidant food. Our research team has been modified and validated to Korean for monitoring dietary quality (References 14 and 15 in the manuscript). In this study, in order to exclude participants having high antioxidant foods, we applied RFS in recruiting participants. We added the brief explain and references in the manuscript (Line 76-77, Page 2).

Why did the investigator remove one participant - did they not meet the inclusion criteria?

: Because of medication, we excluded one participants. Others withdraw the consents. We added this point in the manuscript. (Lines 189-192, Page 5)

In regards to the capsules could you confirm that they contained what you think they did or is there some sort of accreditation you could quote in regards to CKD Bio corp to reassure us they will be of a quality appropriate for your study?

: In order to guarantee the quality of the sample, the company provided test samples checked the levels of heavy metals (lead, total arsenic, cadmium, and total mercury) and coliform group. We have added the above. (Line 94-95, Page 3)

When was the faeces collected? The analytical methods are explained in good detail. I assume the labs were commercial, as stated, or university or clinical hospital?

: Thank you for your valuable comments. Five stool samplers, two toilet papers, and two ice packs were distributed to the study subjects to collect feces. Stool samples were collected at weeks 0 and 12. Based on this part, we revised the part of manuscript. (Lines 107-109, Page 3). We used all commercial, hospital and university labs. Hospital lab collected feces from participants, microbial community was analyzed in the commercial lab and other metabolites in feces were analyzed in the university lab.

Are the labs accredited e.g. according to ISO 15189?

: For some biochemical markers, analyses were done by ISO 15189 certificated lab and other biochemical markers were analyzed hospital lab which has been certificated Good Clinical Practice hospital by Korean government. Other metabolites and microbial community were analyzed in general lab. We clarified ISO certification in the manuscript. (Line111-112, Page 3)

I am not a statistical expert but in reading about all the methods done I wonder if there are enough samples to justify so many analyses and comparisons?

: Thank you for your valuable opinion. In this study, the sample size was determined by power calculation like to other clinical trial. The sample size was estimated to be 35 subjects per group to provide an 80% power of demonstrating a significant difference in TG levels, based on similar previous study and a drop-out rate of 20%. We added the power calculation in the manuscript. (Line 171-173, Page 5)

The table of differences shows those changes that were significant. In some cases the degree of rise wasn't much e.g. LDL. Why were the values increasing so much in the placebo group?

: Thank you for your valuable opinion. As we stated in the question 1, the current study was designed to monitoring postprandial responses for lipid parameters after LPQ180 supplementation. LPQ180 was developed to use as a food supplement or functional food ingredient. Therefore, because of large individual variations, most fating lipid parameters did not show much differences. Most fating values in placebo showed similar pattern to LPQ180 group, however, as postprandial values might be a responsive value for taking high lipid formula and test sample and LPQ180 was effective for inhibitory activity of lipase, the relatively large differences might be shown. We revised this point in the results and discussion, clearly. (Lines 207-210, Page 6; Line 298, 304-305, 309-310, Page 10)

Was the placebo tablet making lipids worse?

: As you see in Table 2, placebo group showed increasing pattern after OLTT although no statistical significance. We mentioned this point in the results. (Lines 212-214, Page 6)

In looking at the changes one thinks about whether the changes are clinically significant, which brings us back to the power calculation, what sort of change in triglyceride would you need to cause to have a meaningful effect on CVD?

: Thank you for your valuable feedback. In daily life for healthy population, maintaining postprandial TG levels might be important. Therefore, in this perspective, we suggested next clinical trial might be needed focused on postprandial TG levels considering baseline gut environment. (Line 373-376, Page 11)

I am not sure we actually know that but it seems the effect is likely less than that required particularly if you account for biological and analytical variation. Microbial analysis is not my area of expertise. On Figure 2 I am afraid I don't fully understand. What do V2 and V4 and the colors in the 2 charts represent? Is this something to do with the primers used to analyse?

: The white bar represents the second visit (Week 0), and the gray bar represents the fourth visit (Week 12). Based on your comments, we have modified figure 2. (Figure 2, Page 8)

In order to analyze microbial community, 16S V3-V4 regions were analyzed using primers. We already described this method and added brief explain for Shanon index and NMDS in discussion (Line 336-339; Page 11)

You also talk about significantly different here but the p value is non-significant isn't it? I am now also a bit confused when you refer to table S1 - where is that (I have 3 supplemental figures but no table). The p value for Indole isn't significant and p-cresol is higher than the 0.019 you quote in the table for indoles and phenols together. I may be getting confused though. The number of comparisons again I find slightly concerning. Looking at the supplemental figures I wonder if those are actually best dispensed with so you can focus on a simple message otherwise I think you need to explain and justify this use of the data better.

: In accordance with your comments, we changed all supplemental figures into tables. the supplemental chart with a supplementary table to avoid confusion. (See Supplementary Table 2 and 3; Line 265 and 277, Page 9)

Also, all the supplementary table was attached to the supplementary compressed file. We corrected the p-values of indole and phenols in the manuscript. (Line 252-255, Page 8)

Does the effect of baseline characteristic on end effect mean that some people you would predict would respond better than others if you were using this therapeutically?

: Thank you for your valuable comment. Based on the current correlation results, population having high levels of enteric bacteria such as R. bromii, K. alysoides, B. intestinihominis and F. plautii or high levels for biogenic amines might be more responsive group for LPQ180 supplementation. In order to prove this hypothesis derived from the current study, other clinical trial might be needed. We added this point in the conclusion part (Line 365-376, Page 11)

does the reduction in indoles etc really prove you have reduced harmful bacteria, should we more circumspect? The discussion certainly discusses much of the literature but some of it I wasn't sure how relevant it was to your actual data and I felt it lacked a critical appraisal of your data.

: Thank you for your valuable comment. Since we discussed for the meaning of indole and phenols in feces, in the current study, we did not catch the significant differences in the levels of harmful bacteria. Therefore, we changed the expression such as harmful or beneficial bacteria into gut environment throughout the manuscript (Line 28-30, Page 1; Line 344-345, 357-361 and 365-376, Page 11)

Reviewer 2 Report

The authors performed an RCT to evaluate the impact of 12-week supplementaiton with L. plantarun Q180 or placebo on blood lipid levels and gut microbiota.  The paper is written very well, but I worry that the authors have overinterpreted the results.

The authors conclude that Q180 may help to prevent hypertriglyceridemia by improving fasting blood lipid levels.  Yet, in Table 2, under Fasting Lipids, the bulk of the lipid variables are no different than placebo.  Further, the variables that are statistically different have little clinical importance.  For example, the change in LDL was about 10 mg/dl different groups, hardly a difference that would be considered meaningful.

I have the same comments regarding the bulk of the results in the paper.  That is, an over-reliance on whether the p-value is less than 0.05, but little in the way of actual interpreting whether the results have any clinical relevance.

Also, the study recruited individuals with triglyceride (TG) levels under 200  mg/dl, yet there was no lower threshold.  If a subject presented with TG=100 mg/dl, why would this subject be considered for the study when the TG is not elevated and the potential for improvement is negligible.

Other minor comments include: Lines 92-94 are Results, not Methods.  Also, in Table 1, for baseline values, SD should be used as the marker of variation instead of SE.

Author Response

Thank you for your valuable comments. We have revised the manuscript to reflect all the comments in the reviewer.

The authors conclude that Q180 may help to prevent hypertriglyceridemia by improving fasting blood lipid levels. Yet, in Table 2, under Fasting Lipids, the bulk of the lipid variables are no different than placebo. Further, the variables that are statistically different have little clinical importance. For example, the change in LDL was about 10 mg/dl different groups, hardly a difference that would be considered meaningful.

: Thank you for your valuable feedback. The current study was designed to monitoring postprandial responses for lipid parameters after LPQ180 supplementation. LPQ180 was screened using inhibitory activity of porcine pancreatic lipase and developed to use as a food supplement or functional food ingredient. Therefore, because of ethical and regulatory issue, we included relatively healthy populations. In this perspective, we revised our manuscript and title focused on the postprandial responses. (Title; Line 19-20, Page 1; Line 64-68, Page 2)

I have the same comments regarding the bulk of the results in the paper. That is, an over-reliance on whether the p-value is less than 0.05, but little in the way of actual interpreting whether the results have any clinical relevance.

: Thank you for your valuable feedback. As we discussed in the question 1, because our participants were relatively healthy populations, it was difficult to catch the clinical relevance using fating values. Instead of fasting values, the current study has been focused on postprandial values, we clearly stated this point throughout the manuscript. (Title; Line 19-20, Page 1; Line 64-68, Page 2; Lines 207-210, Page 6; Line 298, 304-305, 309-310, Page 10; Line 365-376, Page 11)

Also, the study recruited individuals with triglyceride (TG) levels under 200 mg/dl, yet there was no lower threshold. If a subject presented with TG=100 mg/dl, why would this subject be considered for the study when the TG is not elevated and the potential for improvement is negligible.

: Thank you for your valuable feedback. As we discussed question 1, because LPQ180 was screened using inhibitory activity of porcine pancreatic lipase and developed to use as a food supplement or functional food ingredient, we recruited relatively healthy population and tried to exclude the participants in the scope of medication. We clearly stated this point throughout the manuscript. Also, this might be our big limitation, therefore, we also stated this limitation in the end of manuscript (Title; Line 19-20, Page 1; Line 64-68, Page 2; Line 369-376, Page 11)

Lines 92-94 are Results, not Methods.

: Thank you for your valuable feedback. Based on your comments, we moved that part to the result. (189-192 lines, Page 5)

Also, in Table 1, for baseline values, SD should be used as the marker of variation instead of SE.

: Thank you for your valuable feedback. Based on your comments, we changed the SE value to the SD value. (Table 1, Page 5 and 6)

Round 2

Reviewer 2 Report

The authors have made insufficient changes to the paper since their overall conclusions remain the same.